# Conditional Reprogramming Modeling of Bladder Cancer for Clinical Translation

**DOI:** 10.3390/cells12131714

**Published:** 2023-06-24

**Authors:** Danyal Daneshdoust, Ming Yin, Mingjue Luo, Debasish Sundi, Yongjun Dang, Cheryl Lee, Jenny Li, Xuefeng Liu

**Affiliations:** 1Comprehensive Cancer Center, Ohio State University, Columbus, OH 43210, USAluo.379@osu.edu (M.L.);; 2Department of Medicine, Wexner Medical Center, Ohio State University, Columbus, OH 43210, USA; 3Department of Urology, Wexner Medical Center, Ohio State University, Columbus, OH 43210, USA; 4Center for Novel Target and Therapeutic Intervention, Chongqing Medical University, Chongqing 400016, China; 5Departments of Pathology, Urology and Radiation Oncology, Wexner Medical Center, Ohio State University, Columbus, OH 43210, USA

**Keywords:** conditionally reprogrammed cells, bladder cancer, biomarkers, precision medicine, urine biomarkers

## Abstract

The use of advanced preclinical models has become increasingly important in drug development. This is particularly relevant in bladder cancer, where the global burden of disease is quite high based on prevalence and a relatively high rate of lethality. Predictive tools to select patients who will be responsive to invasive or morbid therapies (chemotherapy, radiotherapy, immunotherapy, and/or surgery) are largely absent. Patient-derived and clinically relevant models including patient-derived xenografts (PDX), organoids, and conditional reprogramming (CR) of cell cultures efficiently generate numerous models and are being used in both basic and translational cancer biology. These CR cells (CRCs) can be reprogrammed to maintain a highly proliferative state and reproduce the genomic and histological characteristics of the parental tissue. Therefore, CR technology may be a clinically relevant model to test and predict drug sensitivity, conduct gene profile analysis and xenograft research, and undertake personalized medicine. This review discusses studies that have utilized CR technology to conduct bladder cancer research.

## 1. Bladder Cancer and Clinical Challenges

Bladder cancer (BC) is among the most prevalent cancers in the world, with more than 540,000 new cases and approximately 100,000 deaths worldwide [1]. In the United States alone, more than 82,000 new cases and 16,700 deaths are reported annually [2]. Among the top 10 cancers, BC is the most underfunded cancer by the US National Cancer Institute based on funding dollars and ratio of funding to mortality [3,4].

Bladder cancer manifests in three major forms: as non-muscle-invasive bladder cancer (NMIBC), muscle-invasive bladder cancer (MIBC), and metastatic bladder cancer.

Approximately 75% of BC cases are NMIBC at diagnosis and are usually treated with transurethral resection (TUR) followed by intravesical instillation of therapeutic agents in high-risk patients.

However, over 60% of patients experience disease recurrences in two years, with some advancing to higher stages. Frequent cystoscopy procedures were performed every few months for disease surveillance and are invasive, uncomfortable, and expensive, making BC the costliest cancer (per case among all cancer types). An assay with high sensitivity to isolate cancer cells from urine would decrease the need for such frequent invasive and expensive surveillance cystoscopies.

About 20% of BC patients were MIBC at diagnosis, which requires surgery plus systemic therapy or radiation plus systemic therapy for a curative intent. Neoadjuvant cisplatin-based chemotherapy is the standard of care to downstage tumors prior to surgery and was proven to improve overall survival, compared with surgery alone. These patients have a 5-year survival of 50% [5]. In contrast, patients with residual disease after neoadjuvant therapy have a survival rate of only 35%. Imaging tests and cystoscopy are routinely performed to assess the efficacy of neoadjuvant therapy. For patients who receive radiation therapy, surgery is still indicated if there is a lack of response to radiation. At current stage of disease regardless of treatment, 5-year overall survival is merely 50% percent [6]. Despite the rapid progress in molecular identification and therapeutic targeting of many types of cancer, bladder cancer has lagged behind [7,8]. DNA profiling has revealed tumor heterogeneity that likely leads to variable therapeutic response [9], necessitating preclinical models that can recapitulate in vivo tumor phenotypes.

For metastatic BC, first-line chemotherapy with GC (gemcitabine plus cisplatin/carboplatin) and MVAC (methotrexate, vinblastine, doxorubicin/Adriamycin, and cisplatin) is commonly used and produces a response rate of around 45% [10]. Immunotherapy with pembrolizumab can be used upfront if patients are not platinum-eligible, with a response rate of approximately 27% [11,12,13], while several PD-1 and PD-L1 inhibitors can be used after platinum-treatment failure, with a response rate of about 20% [11,12,13].

Erdafitinib, a targeted tyrosine kinase inhibitor against FGFR1-4, was approved in 2019 based on a response rate of 40% for metastatic urothelial carcinoma with an FGFR3 mutation or FGFR2/3 fusion, which occurs in about 20% of advanced urothelial carcinomas [14,15]. Recently the antibody–drug conjugates enfortumab vedotin and sacituzumab govitecan have also received accelerated approval in the third line and beyond setting [16,17]. Despite recent approvals of new drugs, only a small subset of patients achieve long-term remission, while the majority of patients eventually develop treatment resistance and die from the disease. If a preclinical model is available to recapitulate in vivo patient cancer response and resistance, it can potentially develop novel therapies to overcome resistance and improve patient care. Clearly there is a critical unmet need for a robust method to monitor and predict drug efficacy and potential recurrence in patients with BC.

## 2. Modeling Patient’s Bladder Cancer and Development of CR Technology

Traditionally, cancer research and drug development utilize cancer cell lines [18,19,20,21,22] and genetically engineered mouse models (GEMM) [23]. Traditional established cell lines, derived from human tumors and cultured in two-dimensional or animal models, have been broadly used as bladder cancer models [24]. Even though these models play important roles in bladder cancer research, they have numerous limitations. Only up to 10% of cell lines (depending on the origin tissue) can be successfully established to be transformed immortalized or tumorous cell lines, and most primary cells, particularly normal cells, are hard to culture due to their short life span [25]. Additionally, the genes (DNA sequences) of the established cell lines have undergone remarkable changes, making it challenging to fully recapitulate the complex characteristics of the primary tissue. Therefore, marked differences between the in vivo tumor microenvironment and the in vitro culture increases worries that these cell lines do not accurately mimic tumors in the patients [26]. Animal models, especially recently emerged engineered mouse models and primary patient-derived xenografts, may overcome the restrictions of cell lines and better imitate human disease and therapeutic response [27]. However, the application of animal models might be restricted by low throughput, high costs, and technical difficulties; in addition, differences in specific species cause incorrect recapitulation of biological and therapeutic responses [28]. In recognition of the many limitations of these cancer models, patient-derived models of cancer (PDMCs) were developed recently, including patient-derived xenografts (PDXs), CRCs, and organoids.

### 2.1. Patient-Derived Xenografts (PDX)

In recent years, PDX models faithfully recapitulated the original patient genetic profile, gene expression patterns and tissue histology [29,30] and are widely recognized as a more physiologically relevant preclinical model. Despite their benefits, PDX models are limited by their inherent variability, lower throughput, and lack of growth in vitro. Development of stable PDX lines remains a challenge due to murine stromal outgrowth, lineage commitment, and limited differentiation potential. Overall, it takes 2–5 months for PDX expansion at a relatively high cost for mice and their care.

### 2.2. Organoid Cultures

The laboratories of Clevers and others successfully initially established organoid cultures from mouse tissues and subsequently from human specimens [31,32,33,34,35,36]. Organoid cultures provide a platform to investigate basic biology for cancers, identify drug targets, and study drug resistance [37,38,39]. In vitro three-dimensional (3D) organoid culture systems for different cell types have been successfully established [40]. Mullenders et al. developed a culture system of human bladder cancer cells. With this system, human bladder cancer organoid cultures can be efficiently obtained from resected tumors and biopsies and can be cultured for a long time [41]. In vitro 3D organoids have advantages in understanding the biological processes and mechanisms of bladder cancer. Although the 3D organoids constructed in vitro already have the structure of some organs compared to the intact organs in vivo, the structure is still relatively simple, so it can only partially reflect the tissue characteristics [42]. Organoid cultures are established using 3D growth of epithelial cells in Matrigel™, providing an opportunity for its application to personalized and regenerative medicine (Table 1). Challenges include a requirement for precise tumor tissue sampling for organoid cultures, as this culture system propagates both normal and cancer cells. Overall, it takes 4–6 weeks to provide enough cells for low-throughput drug screening.

### 2.3. Conditionally Reprogrammed Cells (CRCs)

It remains a challenge to establish from a variety of clinical samples (surgical specimens, needle biopsies, cryopreserved tissues, blood, and urine) a single model system that is rapid and simple to perform and has a high rate of success. Recently we discovered a new primary cell culture system, called conditional reprogramming. Using this method, we added irradiated Swiss-3T3-J2 mouse fibroblast cells and Y27632, a Rho-associated kinase (ROCK) inhibitor, to the cell culture plate to efficiently and rapidly produce infinite cells (Figure 1) [43]. Cells generated by this technology are called conditionally reprogrammed cells (CRCs). CR technology can efficiently produce large numbers of primary cells from various tissues, such as fine-needle aspiration, core biopsies, surgical specimens, and patient-derived xenograft tissues [44]. CRCs can be reprogrammed to maintain a highly proliferative state and recapitulate the genomic characteristics and histological characteristics of the primary tissue [45]. Furthermore, once these conditions are removed, the phenotype is fully reversible [46]. Therefore, CR technology might be an ideal model to study bladder cancer, perform gene profile analysis, and test drug sensitivity and for use in regenerative medicine and xenograft research. An important feature is that these cultures can be used for establishing xenografts [47], patient-derived xenograft cell lines [48], cell cultures from PDXs, and organoid cultures. Finally, conditionally reprogrammed cells retain cell lineage commitment and maintain the heterogeneity of cells present in a biopsy [43,47,49,50,51,52]. The reported benefits of CR include ease of execution, exponential growth, a high success rate, and genotype stability in a single model system. These characteristics qualified CR as an outstanding in vitro model compared to other models, such as organoids and patient-derived xenograft [44]. Applications of CR technology in clinical research have been studied in prostate [49], bladder [53], breast [54], lung [55], liver [56], and gastric [57] cancers. The CR system is applicable to many epithelial tissues including prostate, kidney, skin, and lung, among many others [45]. Furthermore, usage of the system is not restricted to humans; it can be used on many mammalian species such as rats, mice, dogs, horses, and cows [58]. A remarkable feature is that CR cultures can be used for generating xenografts [59] and patient-derived xenograft cell lines [48] and can also be used to establish cell cultures from organoid cultures and patient-derived xenografts. In addition, conditionally reprogrammed cells maintain cell lineage commitment and retain the heterogeneity of cells existing in the original biopsy [43,49,59]. CRC is the only method that can culture tumor cells from tissue, blood, and urine samples of BC patients [41,53,60,61,62,63,64,65]. In this article, we review studies that use CR technology for bladder cancer research.

## 3. CR Cells from Murine Models of Bladder Cancer

High-grade bladder cancer contains intrinsic molecular subtypes and can be divided into luminal-like and basal-like subtypes. Saito R. et al. described the first subtype-specific murine models of bladder cancer and show that Upk3a-CreERT2, Trp53L/L, PtenL/L, Rosa26LSL-Luc (UPPL, luminal-like), and BBN (basal-like) tumors are more faithful to human bladder cancer than the widely used MB49 [66]. Saito R. et al. also utilized the CR technique to generate cell lines from adoptive transfer models for luminal-like UPPL tumors and basal tumors derived from BBN-treated animals. CR cells derived from tumors using the UPPL model maintain luminal-like characteristics, with high expression of Pparg and Gata3 gene signatures. Moreover, gene expression profiles from BBN and UPPL models more closely map to human bladder cancer. These models can be used to interrogate subtype-specific responses to immune checkpoint inhibition and other immunotherapy strategies in vivo. Currently, very few cell lines exist for modeling bladder cancer in immunocompetent mice. MB49 cells are the commonly used syngeneic bladder cancer cell lines [66]. Given the long latency of tumor formation in the UPPL model, this group established tumor cell lines from both UPPL and BBN tumors using the conditional reprogramming of cells (CRC) method [59]. Specifically, transplantable cell lines were established from BBN (BBN963) and UPPL (UPPL1541) tumors and have been confirmed to grow in C57BL/6 mice. Thus, CR cells may couple with animal models for biology and therapeutic studies of bladder cancer.

## 4. CR Cells from Bladder Cancer Tissues

Precision medicine, also known as personalized medicine, is a newly developed strategy related to the biologic characterization of cancers and is considered a new era in cancer prevention and therapy [67]. The applications of precision medicine are comprehensive, including prevention, diagnosis, prognosis, monitoring of treatment response, and early detection of treatment resistance. The purpose of precision medicine is to erase the “one size fits all” model of cancer patient management [68]. Lately, the classification and treatment of various tumors have changed because of genetic assessment. Targeted therapy is a treatment strategy that uses agents to target particular proteins and genes associated with the growth and survival of tumor cells, and this is the base of precision medicine. Due to the lack of appropriate in vitro models for bladder cancer, this is a major concern in studying treatment response and drug resistance. A specific restriction in recognizing effective drugs and targets for bladder cancer is that the results are merely based on studies in long-term xenograft models or cultured cell lines; thus, the consequences of most clinical research are usually unsatisfactory.

At present, the use of patient-derived models (PDMs) with the features of maintained genotype, high immortality, throughput screening, and xenotransplantation is urgently needed for drug screening, drug discovery, and targeted therapy. The CR method can be used in primary cell cultures from normal and tumor tissues of different types of tissues and to retain the genotypic and phenotypic characteristics and heterogeneity of the primary species. CR cells can also be used in 3D conditions and PDXs. Therefore, CR technology offers a new tool for assessing the toxicity and effectiveness of efficient and new drugs and developing personalized treatment strategies in bladder cancer.

Recently, Kettunen et al. investigated the feasibility of CR technology for the characterization of bladder cancer properties and their suitability for personalized drug-sensitivity screening. They used the CR method to establish patient-derived cell cultures from six bladder cancer patients undergoing transurethral resection or radical cystectomy. Four cases were classified as high-grade urothelial carcinoma (pTaN0, pT1N0, pT1N0, pT4aN1), one case was diagnosed as small-cell neuroendocrine carcinoma (pT4aN1), and one case had primary bladder adenocarcinoma (pT2bN1). Kettunen et al. established cultures based on the Liu et al. procedure [59]. Four (67%, a small-cell carcinoma and three urothelial carcinomas) out of six were successfully generated while cultured for five passages and repropagated after cryopreservation for later analysis. Even though all four CR cultures revealed similar morphology compared to the primary specimens, exome sequencing showed that only two of them maintained the majority of genetic mutations, such as RB1 mutation, detected in respective corresponding tumors (small-cell carcinoma and pT1N0 urothelial carcinoma). In immunohistochemical staining analysis, urothelial carcinoma (pT1N0) CR cells revealed strong cytokeratin 5/6 expression, which indicates a shift to a basaloid phenotype. Similar to the primary tumor, loss of cytokeratin and strong expression of neuroendocrine biomarkers have been shown in small-cell carcinoma CR cells. In comparison to noncancerous cultures, both cancerous cultures were highly p53-positive and indicated a higher proportion of Ki-67-positive cells. In the other two cultures that could not maintain the specific driver alterations, the investigators proposed contamination of noncancerous cells as the potential cause. Thus, careful genomic analysis and selection of parental tumor material are critical in confirming the source of the established culture. In vitro drug-sensitivity screening assessment in the two cancerous CR cultures demonstrated sensitivity to cisplatin, taxanes, and gemcitabine, proteasome, and topoisomerase inhibitors. The small-cell carcinoma CR culture also revealed sensitivity to statins [53].

For many years, different methods have been employed to assess personalized drug sensitivity in cancer patients through antimicrobial susceptibility testing. However, no laboratory-based commercial assays are currently available for use in oncology practice due to the greater complexity of tumor genomes and the variability of tumor clones compared to microorganisms. Further investigation is required to advance the development of tests that possess high reproducibility and offer valuable insights for drug selection. These tests should be resilient against the influence of tumor heterogeneity while also being efficient in terms of time and cost. Kettunen et al.’s investigation shows promise for meeting these requirements. Although Kettunen et al.’s [53] method represents an improvement over traditional approaches such as patient-derived xenograft models, it is uncertain whether this new approach can accurately inform the selection of anticancer drugs for individual bladder cancer patients due to its relatively low success rate (two out of six cases, or 33%) and the absence of definitive clinical correlations in the article [68]. Nevertheless, the study serves as a pilot that demonstrates the potential of CR techniques for custom drug-sensitivity screening in bladder cancer.

## 5. Liquid Biopsies and Urine CRCs in Bladder Cancer and Immunotherapies

### 5.1. Circulating Tumor Cells (CTCs): Importance and Technical Challenges

Liquid biopsies are noninvasive methods that may be harnessed for cancer precision medicine. Circulating factors, including circulating tumor cells (CTCs), cell-free DNA (cfDNA), RNAs (miRNAs, long noncoding RNAs [lncRNAs], mRNAs), cell-free proteins, peptides, and exosomes are derived from cells in human body liquids. CTCs from BC can be detected in the urine and serum of patients with metastatic BC, with higher levels of CTCs correlating with cancer aggressiveness [69,70]. CTCs derived from BC can be measured by using CTC-specific proteins, such as c-MET and PD-L1 [71,72,73]. Increased CTC levels were also able to predict clinical outcomes, such as recurrence and survival [74]. CellSearch^TM^, an FDA-approved CTC assay kit, is currently being used in clinical settings for prognostic purposes in breast cancer but has low sensitivity and specificity [75]. Being able to culture cancer cells from blood and other liquid biopsies would be extremely valuable, but there are many technical issues that limit the study of CTC biology and their applications. It is technically difficult to capture and characterize viable CTCs [67,76,77].

### 5.2. Noncirculating Cell Biomarkers:

In addition to CTCs, DNA, RNA, and exosome candidate biomarkers in serum and/or urine have been identified in BC. Mutation of the human telomerase reverse transcriptase gene (hTERT) promoter occurs in 70% of human BC and is also associated with recurrence and poor prognosis of BC [78,79,80,81]. In the urine specimens of BC patients, the telomerase reverse transcriptase (TERT) promoter mutations correlated with recurrence [82], while KRAS2 mutations were found in the plasma even before BC diagnosis [83]. Urinary UBE2C and hTERT mRNA were found to be potential markers for early diagnosis and prognosis of BC [84]. Urinary levels of miR-126 and miR-146a-5p were also found to be elevated in BC and are associated with tumor grade and invasiveness [85]. Exosomes transfer biologically active molecules and can be secreted into the urine, blood, and other body fluids [86]. Hence, exosomes are essential mediators of cell–cell communication [87]. Interestingly, there is a strong association between heightened exosome levels and BC [88]. In urinary exosomes, significantly increased levels of active molecules (e.g., TACSTD2, lncRNAs–HOTAIR, HOX- AS-2, ANRIL, and linc-ROR) were found in high-grade MIBC patients [89]. However, none of the above markers are found in a majority of BC and urine samples of patients and thus have not been suggested for clinic use. Further, they cannot be used for functional analyses to predict clinical outcome.

### 5.3. Urine CRCs as a Living and Functional Biomarker for Clinical Relevance and Response

Our previous work evaluated drug responses via CR culture of bladder cancer cells, which was taken from urine samples. The success rate of urine CRC cultures was high, 83.3% (50 of 60 cases); specifically, low-grade bladder cancer was 75.0% (9 of 12 cases), and high-grade bladder cancer was 85.4% (41 of 48 cases). Interestingly, we obtained a 100% success rate of BC cell cultures after we optimized collection conditions. Additionally, we did not observe bias for the success rates of CRC urine cultures with respect to pathology group, disease status, age, or gender from all bladder cancer cases. We also reported that CR cells had 79.7–82.6% of genetic variation similar to the original tumors. Thirteen patients underwent drug-sensitivity tests, which revealed varied responses to conventional drugs such as gemcitabine, cisplatin, pirarubicin, and epirubicin [62]. In order to study whether urine BC cells can predict clinical response, we collected urine samples before and after treatment. Thus, we were able to establish urine CR BC cells from Patients 1 and 2 before treatment (surgery and/or intravenous chemotherapy, gemcitabine + cisplatin; Figure 2) and from urine samples after treatment; however, we only established UCCC from Patient 1, not Patient 2, after two cycles of chemotherapy. This outcome was consistent with clinical follow-up showing that Patient 1′s tumor soon relapsed after treatment. However, the tumor in Patient 2 showed no recurrence after the first and second cycles of chemotherapy, correlating with our urine BC cell culture results.

Our data also showed that response of CR BC cells generated from patients’ urine samples prior to treatment correlated with response of the BC patients. The CR BC cells of one patient showed relatively low sensitivity to pirarubicin, which was consistent with clinical follow-up. This patient’s tumor soon relapsed 3 months later after intravesical treatment with pirarubicin (40 mg) followed by undergoing transurethral resection. Another patient showed relative sensitivity to gemcitabine and cisplatin. In the absence of surgery, his CT obtained 6 months later showed stable disease (SD) after treatment with gemcitabine (1000 mg/m^2^/intravenous) and cisplatin (70 mg/m^2^/intravenous). These clinical results correlated well with the drug-sensitivity profiles of the corresponding CR cells in response to gemcitabine and cisplatin. Thus, urine BC cells can be used to test drug sensitivity, and their response seemed to correlate with the patients’ response.

### 5.4. Urine CRCs with IFN-Induced PD-L1 Levels as a Novel Biomarker for BC Immunotherapies

Checkpoint blockade immunotherapy has been widely used in bladder cancer treatment, but its effectiveness cannot be determined by traditional preclinical models. For example, increased evidence from both preclinical and clinical studies mostly indicates that the immunochemistry (IHC) of programmed-death ligand 1 (PD-L1) levels in tumor tissues currently used clinically is not a satisfactory predictor of anti-PD-1/PD-L1 treatment outcomes [90]. Glycosylation of PD-L1 may render its polypeptide antigens inaccessible to PD-L1 antibodies, leading to inaccurate IHC readouts in some patient samples [86,91]. Recently we found that different BC cells have different PD-L1 expression and sensitivity to IFN-γ, and these diversities can be found not only between tumor cells from different patients but also among subclones in tumor cells from the same patient. We investigated PD-L1 as a means to predict the immune checkpoint inhibitors response in bladder cancer by CR technology. We collected 43 tumor/urine (tumor = 17, urine = 26) samples of primary bladder cancer patients and generated bladder cancer cells. The expression levels of PD-L1 on bladder cancer cells were measured by flow cytometry before and after interferon gamma treatment. We found that IFN-γ-stimulated PD-L1 (sPD-L1) expression on BC cells may predict the prognosis of BC patients. The better prognostic value was in urine BC-PD-L1. Transcriptome analysis showed that BC cells with high sPD-L1 tended to enrich genes associated with the collagen-containing extracellular matrix, cell–cell adhesion, and positive regulation of the immune system. Urine BC-PD-L1 also exhibited predictive value for ICI response in BC patients. The results showed great potential of urine CR bladder cancer cell PD-L1 for prognostic and predictive value in clinical practice [92]. This would be the first functional predictive biomarker for BC immunotherapy.

Currently, cystoscopy is the gold standard method for diagnosis and surveillance in bladder cancer. It is an invasive, painful, and expensive procedure. Most invasive procedure have complications and possible negative consequences for patients [93]. Therefore, discovering a noninvasive method will be beneficial for diagnosis, drug-sensitivity assessment, and follow-up. Liquid biopsies recently came into use in the clinical setting, and studies have confirmed their ability to reveal various genetic alterations of metastatic and primary tumors and provide precise clinical information [94]. The most easily reachable liquid biopsy type for bladder cancer is urine. Urine is easy to access, and liquid biopsies are noninvasive and appropriate for patient follow-up. Liquid biopsies have been broadly explored for diagnosis and surveillance of bladder cancer by detecting biomarkers such as microRNA, circular RNA, and cell-free DNA [95]. Further studies need to be conducted to determine other possible applications of this noninvasive urine-derived model in bladder cancer.

## 6. Challenges and Future Prospects for Clinical Setting

CR technology has demonstrated its potential to play a vital role in bladder cancer research. However, there are limitations that need to be addressed. For instance, the CR method does not incorporate essential stromal components such as vascular immune cells, matrix elements, and endothelial cells. This limitation hinders the comprehensive analysis of how stromal cells influence tumor cell growth and how tumor cells respond to drugs [27]. Another challenge is distinguishing tumor cells from normal epithelial cells, since sometimes normal cells are generated more than tumor cells such that normal cells may surpass in cultures. However, modifications to the standard CR method can enable selective expansion of tumor cells in vitro [27]. Despite these limitations, CR technology holds excellent application prospects in bladder cancer research. There is no perfect model for biomedical research, and researchers must choose an appropriate model that suits their specific research question. In many cases, scientists use a combination of technologies at different levels, from molecules to cells, organs, and populations, for their research goals. We also would like to highlight several aspects for further development for clinical laboratory setting.

### 6.1. Development of Next-Generation CR Culture System for Urine Cancer Cells: Ready-to-Go Kits

In spite of a 100% success rate of urine CRC cultures in the experimental lab setting, we need to optimize/simplify culture conditions and establish an SOP for future clinical utility in order to further reduce variability of the results in the clinical lab setting. We will be able to establish a series of culture kits and SOPs with a coculture system, a conditioned medium (CM), and hypoxic conditions for maximal efficiency and robustness.

### 6.2. A Novel Strategy and SOP for Maintenance of Urine Cell Viability That Directly Determines the Success Rate of Urine CRC Cultures

Analytical quantification of clinically relevant biomarkers is affected by variation in biospecimen collection, processing, and storage procedures. Urine sample conditions greatly affect cell viability, which directly determines the success rate of urine CRC cultures. We need to provide an SOP for urine sample handling and storage.

### 6.3. Generation of an Easy Urine Self-Sampling SOP/kit for Urine Cancer Cells

Urine samples are completely noninvasive and easy to collect; however, many variables surrounding sample collection might affect cell viability and the success rate of urine CRCs. We need to evaluate how the timing of urine sample collection (days before/after cystoscopy, early morning/noon/evening, etc.) and urine volume (drinking water before collection) affect cell viability and urine CRC cultures. Finally, we will optimize a self-sampling procedure for patients with BC.

## 7. Conclusions

The emergence of CR technology presents promising prospects for researching bladder cancer. By utilizing CR techniques, cell cultures can be rapidly and effectively generated from both normal and cancerous tissues. What is particularly significant about these CR cells is that they preserve the developmental properties of the parental tissue and can regain the capacity for cellular differentiation even after removing certain conditions. Moreover, CR technology can rapidly produce cultures from small biopsy samples and even cryopreserved tissues as well as from xenografts and organoid tissues. This allows for the creation of cell-derived xenograft tumors and the cultivation of spheroids and organoids, making CR technology a potentially optimal in vitro model for bladder cancer research that can facilitate the progress of precision medicine and drug discovery. Moreover, in the future CR technology could help develop personalized medicine and could create a living biobank for bladder cancer. In short, the use of CR technology provides exceptional opportunities for advancing the diagnosis, treatment, and prevention of bladder cancer.

## Figures and Tables

**Figure 1 cells-12-01714-f001:**
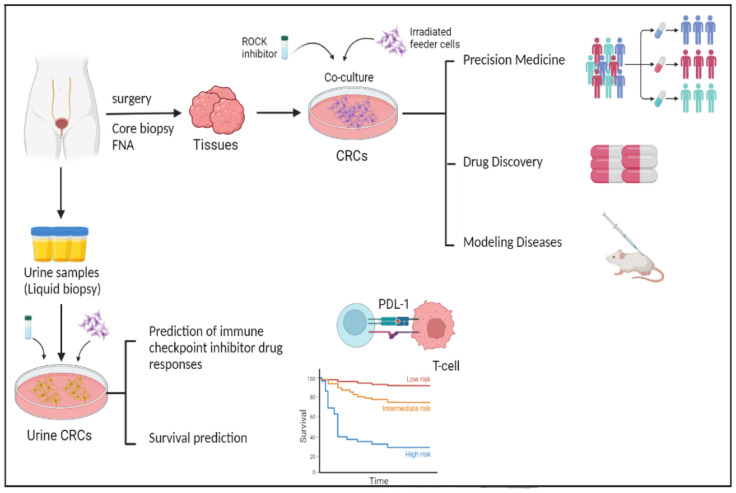
CR technology applications in bladder cancer. The CR method can quickly generate cultures from normal and cancerous tissue obtained through fine-needle aspiration (FNA), core biopsy, and surgery. Therefore, CR technology can be used as an ideal in vitro model for bladder cancer research especially in precision medicine. This technology can be used to predict immunotherapy drug responses and subsequently survival time. The figure was drawn using BioRender.

**Figure 2 cells-12-01714-f002:**
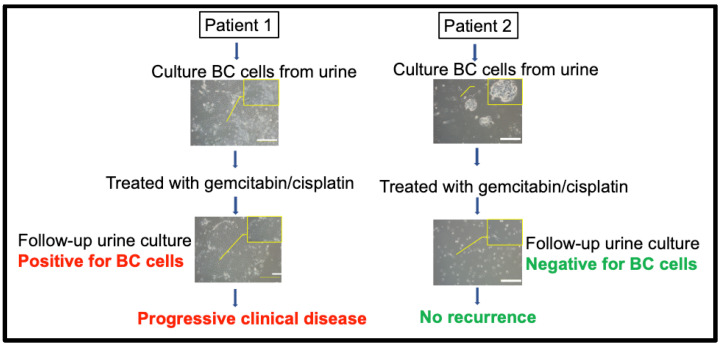
Clinical relevance of urine CR BC cells from Patients 1 and 2. Patient 1 accepted tumor resection following chemotherapy but soon relapsed (red). Patient 2’s tumor was sensitive to chemotherapy. Phase contract images of urine CR cell conditions before and after chemotherapy treatment in Patients 1 and 2. We failed to establish urine CR BC cells after the first and second cycles of chemotherapy in Patient 2 since this patient was disease-free (green). Red color indicated positive CR BC culture and active disease (non-responder, or BC recurrence). Green color indicated negative CR BC culture and disease free (responder, or no recurrence). The figure was drawn using BioRender.

**Table 1 cells-12-01714-t001:** Comparisons between patient-derived models: 2D culture, organoids, patient-derived xenografts (PDXs), and conditionally reprogrammed cells (CRCs).

Methods	Advantages	Disadvantages
2D culture	1. Inexpensive technique 2. Easy to manipulate genetically	1. Loss of tumor heterogeneity 2. Lack of microenvironment
Organoids	1. 3D culturing 2. Can generate both healthy and tumor organoids 3. Maintain tumor heterogeneity	1. Dependent on stem cells 2. Overgrowth of nonmalignant cells
PDXs	1. In vivo model 2. Direct engraftment from human tumor 3. Maintain histological, genomic, and transcriptomic features of tissue of origin 4. Recapitulate the natural environment of the tumor	1. Expensive technique 2. Resource- and time-consuming 3. Not suitable for high-throughput drug screening 4. Rely on interactions with a mouse microenvironment 5. Only tumor models 6. Challenging to be reproducible on a large scale
CRCs	1. A wide range of specimen sources 2. Paired normal and tumor cells culturing 3. Cost savings and rapid expansion (1–10 days) 4. Can maintain original karyotype and tumor heterogeneity 5. High-throughput drug screening	1. Contamination with feeder cells 2. Overgrowth of benign cells 3. Lack of stromal components (matrix elements, vascular immune cells)

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
