# Peer review of "Conditional Reprogramming Modeling of Bladder Cancer for Clinical Translation"

_cells, 2023, doi:10.3390/cells12131714_

Round 1

Reviewer 1 Report

Overall, the review is well-structured and comprehensive, providing good information on various aspects of bladder cancer modeling. However, some sentences required more clarifications:

Line 85-86: "the genes of the established cell lines": is it gene expression or DNA sequence

Line 125-126: The sentence is unclear.

Line 194: "CR cells can also be cultured in 3D conditions and PDMs." CR cells cultured in PDMs is unclear.

Line 226-228: The sentence is unclear.

Line 254-259: The paragraph is unclear.

Line 265-267: The sentence is unclear.

Line 272: "as below" is not explained.

Line 276-279: The paragraph is unclear, and does not match the figure.

Line 306: The "programmed-death ligand 1 (PDL-1)" needs to be addressed earlier in the text and there is inconsistency in use of PDL-1 vs PD-L1.

Line 316-317: The sentence is unclear.

Line 348-349: The sentence is unclear.

Line 351-352: The sentence is unclear.

The quality of the figures is poor, and the figure legends need to be written with greater clarity and better explanations.

As mentioned, it would be helpful if the text corresponding to the aforementioned lines is modified to provide clearer explanations.

Author Response

Reviewer #1                    Comments and Suggestions for Authors

Overall, the review is well-structured and comprehensive, providing good information on various aspects of bladder cancer modeling. However, some sentences required more clarifications:

Line 85-86: "the genes of the established cell lines": is it gene expression or DNA sequence

RE: It is DNA sequence, we corrected this in the text.

Line 125-126: The sentence is unclear.

RE: We corrected as “In this method, we added irradiated Swiss-3T3-J2 mouse fibroblast cells and Y27632, a Rho-associated kinase (ROCK) inhibitor, to the cell culture plate to efficiently and rapidly produce infinite cells”

Line 194: "CR cells can also be cultured in 3D conditions and PDMs." CR cells cultured in PDMs is unclear.

RE: We changed “cultured” to “used” and “PDMs” to “PDXs”.

 Line 226-228: The sentence is unclear.

RE: The sentence is “It is necessary to conduct additional research to develop tests that are both highly reproducible and capable of providing useful information for drug selection, while remaining unaffected by tumor heterogeneity, as well as being time-efficient and cost-effective”. We changed this to “Further investigation is required to advance the development of tests that possess high reproducibility and offer valuable insights for drug selection. These tests should be resilient against the influence of tumor heterogeneity while also being efficient in terms of time and cost”.

Line 254-259: The paragraph is unclear.

RE: We made changes as “In addition to CTCs, DNA, RNA and exosome candidate biomarkers in the serum and/or urine have been identified in BC. Mutation of the human telomerase reverse transcriptase (hTERT) promoter occurs in 70% human BC, and is also associated with recurrence or poor prognosis of BC [84-87]. In the urine specimens of BC patients, the telomerase reverse transcriptase (TERT) promoter mutations correlated with recurrence [88], while KRAS2 mutations were found in the plasma even before BC diagnosis [89]. Urinary UBE2C and hTERT mRNA were found to be potential markers for early diagnosis and prognosis of BC [90]. Urinary levels of miR-126 and miR-146a-5p were also found to be elevated in BC and are associated with tumor grade and invasiveness [91].

Line 265-267: The sentence is unclear.

RE: We changed this to “However, none of the above markers are found in a majority of BC or urine samples of patients, thus  have not been suggested for clinic use.”

Line 272: "as below" is not explained.

RE: We made change as “Interestingly we obtained 100% success rate of BC cell cultures after we optimized collection conditions.”

Line 276-279: The paragraph is unclear, and does not match the figure.

RE: We made changes as “Thirteen patients underwent drug sensitivity tests, which revealed varied responses to conventional drugs like gemcitabine, cisplatin, pirarubicin, and epirubicin. However, no clinical susceptibility data was provided for other drugs: paclitaxel, lapatinib, bortezomib, and docetaxel, which showed significant inhibition of urine culture growth [95]. Thus, urine BC cells can be a promising tool to assess drug responses and sensitivity by a minimal invasive method rather than invasive procedures.”

Line 306: The "programmed-death ligand 1 (PDL-1)" needs to be addressed earlier in the text and there is inconsistency in use of PDL-1 vs PD-L1.

RE: we addressed this issue in the text. All turned to PD-L1.

Line 316-317: The sentence is unclear.

RE: We made change as “The results showed great potential of urine CR bladder cancer cell PD-L1 for prognostic and predictive value in clinical practice”

Line 348-349: The sentence is unclear.

RE: We made changes to “For instance, the CR method does not incorporate essential stromal components like vascular immune cells, matrix elements, or endothelial cells. This limitation hinders the comprehensive analysis of how stromal cells influence tumor cell growth and their response to drugs. “

Line 351-352: The sentence is unclear.

RE: We made changes as “Another challenge is distinguishing tumor cells  from normal epithelial cells, since sometimes normal cells are generated more than tumor cells, so normal cells may surpass in culture.”

The quality of the figures is poor, and the figure legends need to be written with greater clarity and better explanations.

RE: We changed figures with higher resolution and corresponding figure legends. We also added a new Table with comparisons for the current models.

Table 1 | Comparisons between patient-derived models: 2D culture, organoids, patient-derived xenografts (PDXs), and conditionally reprogrammed cells (CRCs)

Methods

Advantages

Disadvantages

2D culture

1. Inexpensive technique

2. Easy to manipulate genetically

1. Loss of tumor heterogeneity

2. Lack of microenvironment

Organoids

1. 3D culturing

2. Can generate both healthy and tumor organoids

3. Maintain tumor heterogeneity

1. Dependent on stem cells

2. Overgrowth of nonmalignant cells

PDXs

1. In vivo model

2. Direct engraftment from human tumor

3. Maintain histological, genomic, and transcriptomic features of tissue of origin

4. Recapitulate the natural environment of the tumor

1. Expensive technique

2. Resource and time consuming

3. Not suitable for high-throughput drug

Screening

4. Rely on interactions with a mouse

Microenvironment

5. Only tumor models

6. challenging to be reproducible on a large scale

CRCs

1. A wide range of specimen sources

2. Paired normal and tumor cells culturing

3. Cost saving and rapid expansion (1-10 days)

4. Can maintain original karyotype and tumor heterogeneity

5. High-throughput drug screening

1. Contamination with feeder cells

2. Overgrowth of benign cells

3. Lack of stromal components (matrix elements, vascular immune cells)

Reviewer 2 Report

Overall, this review has a good structure demonstrating the models available for addressing drug efficacy and disease modeling, however the writing style of this review is a mixture of systematic, and non-systematic review. 

Sometime, authors can be partial is selecting the research articles. There are certain paragraphs in this venison that implies fixed ideas, and bias of the authors with regards to a certain method (ex: lines 130 -136).  

I recommend the authors to view the subject from a large perspective. 

Also, it would be more appropriate to describe the view and state the pros and cons of each method without showing data on one certain method over the others. Please remove figures 1 and 2 since this is not a research article and replace it with the overall understating and the message of the research article, while comparing it with the others. 

Section 5 of the article is nicely done.

English language is fine.

Round 2

Reviewer 2 Report

No Further comments